

# Investigating Los Angeles' urban roadway network from a biologically-formed perspective

Sophia Deen[1], Tatiana Kuzmenko[2], Hossein Asghari[3] and Demian A. Willette[2]

[1] Department of Health and Human Sciences, Loyola Marymount University, Los Angeles, CA, United States of America
[2] Biology Department, Loyola Marymount University, Los Angeles, CA, United States of America
[3] Department of Electrical Engineering and Computer Science, Loyola Marymount University, Los Angeles, CA, United States of America

## ABSTRACT

The evolution of networks is constrained by spatial properties of the environment; a characterization that is true in both biological and built networks. Hence built networks such as urban streets can be compared to biological networks to reveal differences in efficiency and complexity. This study assessed foraging networks created by the slime-mold *Physarium polycephalum* on proportional 3D-printed topographic maps of metropolitan city of Los Angeles, California. Rapidly-generated isomorphic solutions were found to be consistently and statistically shorter than existing roadways in system length. Slime mold also allocated resources to supporting key nodes, analogous to how heavy traffic flows through major intersections. Further, chemical deterrents inhibited exploration of slime mold in selected areas and allows for testing of network redundancy and system resilience, such as after an earthquake or wildfire.

Corresponding author
Demian A. Willette,
demian.willette@lmu.edu

## INTRODUCTION

Networks are created where paths crisscross one another, forming a collection of nodes in space (*Barthelemy, 2011*). Networks are commonly found across biological and built environments to facilitate transport of materials and information. Cardiovascular systems, neural networks, ant tunnels and termite colonies are all examples of biological transport networks, whereas roadways, power grids, sewage systems, and telecommunications are built infrastructure examples. The evolution of a network and resulting topography are both constrained by spatial properties of the environment; a characterization that is true in both biological and built networks. Hence built networks such as urban streets can be compared to biological networks to reveal differences in efficiency and complexity (*Jiang, Zhao & Yin, 2008*). Urban road networks are essential for the movement of goods, people, and services in metropolitan areas, yet these networks were built over many years that span dynamic economic, political, technological and legal periods that did not inherently promote maximum transit efficiency. Economic drivers may discourage redundancy, yet
the availability of multiple pathways allows for the self-healing of networks (*Quattrociocchi, Caldarelli & Scala, 2014*) and ensures network resilience after disruptions (*Tero et al., 2010*).

The ways in which urban road networks are constructed has been examined in an assortment of perspectives including through statistics (*Samaniego & Moses, 2008*) and computational modeling (*Li et al., 2015*). Roadway creating models typically involve advanced mathematical models that may be difficult to interpret (*Yang & Bell, 1998*), which may impede many from truly understanding computed designs, including stakeholders who would benefit from a greater understand of network formation. Even when artfully sketched out designs of proposed networks are created, there are limitations in how closely these models may be compared to real-world networks (*Jiang, 2009*). The mathematically designed models can be a crucial first look at roadways particularly for their accuracy and clarity yet having a more tangible and physical model would be beneficial in other ways such as seeing how the network would interact in its environmental surroundings (*Samaniego & Moses, 2008*). Furthermore, translating urban roadway models created mathematically from two-dimensions to three-dimensions are traditionally costly and time consuming to construct, yet provide valuable insight for comparison with real-world networks. A complementary approach to exploring built infrastructure systems is by using biologically created networks, specifically the network-constructing plasmodial slime mold *Physarium polycephalum* (*Tero et al., 2010*; *Adamatzky & Akl, 2011*; *Adamatzky, 2014*).

*Physarum polycephalum* is an aceullar, multinucleated ameboid organism that can be observed with the naked eye. In its vegetative stage called plasmodium, *P. polycephalum* explores the environment in search for food (decaying organic matter) using chemotaxis and a contiguous distribution strategy to maximize the area covered. The plasmodium can explore its terrain at a rate up to 5 centimeter per hour (*Kessler, Aldrich & Daniel, 1982*). Once food is identified, the plasmodium connects to the food by developing a complex and highly optimized network of pulsating veins or tubes. Plasmodium tubes contain actin and myosin as the components to achieve locomotion, shuttle streaming, and transportation of nutrients and chemical signals within the organism (*Alt, 1985*). It has been shown repeatedly that *Physarium* exhibits intelligent-like decision making and algebra computing abilities while maintaining the plasmodial network (*Nakagaki, Yamada & Toth, 2001*; *Tero et al., 2010*; *Evangelids et al., 2017*). For example, the plasmodial slime mold has the capability of creating networks isomorphic to that of the Tokyo Railway system, which is highly regarded for its efficiency (*Nakagaki, Yamada & Toth, 2001*; *Tero et al., 2010*). This is especially impressive given the plasmodial network develops within a few days without any centralized control, whereas the railroad network required years of design and formation effort from civil engineers. What has been lacking in the majority of studies (*Adamatzky & Jones, 2010*; *Tero et al., 2010*; *Adamatzky, 2014*; *Adamatzky et al., 2014*) on this plasmodium is that they were conducted on the flat surface in the absence of real-world topographic features such as mountains and valleys that have a dramatic impact on the development of the road networks (but see *Adamatzky, 2014*; *Evangelids et al., 2017* as exceptions). Furthermore, slime mold is gravisensitive and positively geotrophic (*Moore & Cogoli, 1996*) meaning in a 3D environment, it will avoid higher elevations

and adjust its network structure to accommodate the landscape (*Evangelids et al., 2017*). This biological feature and an increased accessibility of three-dimensional (3D) printing technology makes it possible to investigate the potentially powerful capacity of slime mold as a tool for exploring and potentially re-envisioning urban roadway networks in the cities where prominent geographical features have a strong impact on transportations network development.

Here we examine how the biological network-building *P. polycephalum* explores the major metropolitan city of Los Angeles, California and compare it to the existing system of freeways. Our study focuses on the downtown Los Angeles region, an area nestled between mountains and foothills, flat basin, and the Los Angeles River, spanning an elevation of 50–500 m above sea level. The urban roadways and associated traffic patterns of Los Angeles have been the focus of study for decades (*Stonex, 1957*; *Teague et al., 1972*; *Boarnet, Kim & Parkany, 1998*) and continue to encourage novel thinking in solving congestion and reducing commute times (*Wang et al., 2018*; *Zhou, Murphy & Corcoran, 2018*; *Gonzalez, 2019*). As the United States' second largest city and among the most densely populated (*U.S. Census Bureau, 2018*), Los Angeles is widely known for its extensive roadways and congested traffic. Seemingly paradoxically, the combination of Los Angeles' spatial extent and longer road segments contribute to its ranking as the top US city for accessibility to jobs by automobile (*Levinson, 2012*), with higher job accessibility statistically correlated with shorter commute times in the city (*Hu, 2015*).

This study specifically aims to assess the exploration of the slime mold's foraging on proportional geographic locations of 3D printed topographical maps of Los Angeles. The results from *P. polycephalum*'s exploration will be used to assess the slime mold's ability to generate isomorphic solutions similar to Los Angeles' existing roadway. Slime mold and real-world roadway lengths will be calculated and contrasted to identify which are shorter, thus identify a potentially more efficient network system. This study is not the first to attempt to combine emerging 3D printing technology with network-solving capability of *P. polycephalum* (see *Evangelids et al., 2017*), yet to our knowledge, it is the first to couple the two to examine applied network design at the scale of a single metropolitan city.

## MATERIALS & METHODS

Proportional topographic maps of a 12 km by 15 km area of downtown Los Angeles were captured from Google Earth (Alphabet, California), edited using the Blender (*Blender Foundation, 2018*) and MakerBot (Stratasys, NY, USA) software packages and converted into stereolithography files (.stl) using .stl generator software, and printed on a Makerbot 1st Generation Replicator. To have a measurable effect of plasmodium behavior all the elevation values were enhanced 2.5 times. 12 cm × 15 cm printed maps (1:100,000 scale) were marked with a dot of permanent paint in fixed locations where the food source (oat flakes) could be placed identically through replicated trials. Seven food source locations were chosen intentionally to align with major intersections of both highways and major roadways in Downtown Los Angeles.

For each trial each 3D topographic map was placed in a large 20 × 20 cm square petri dish containing an absorbent towel soaked in deionized water. The map, dish, and towel

were sanitized for one hour under a UV light to inhibit any fungal growth or contamination. After sanitization, a thin coat of autoclaved non-nutritive 2% agar was painted over the surface of the 3D maps. The agar media provided a superior growing surface to bare plastic yet was sufficiently thin enough to not distort scaling of the 3D maps. Agar was allowed to solidify and was then ready for the placement of slime mold and oat flakes at fixed starting locations.

Plasmodial *P. polycephalum* was cultured into its active form from purchased dry sclerotium (Carolina Biological Supply Company, North Carolina), slime mold's dormant stage. The dry sclerotium was placed in the center of a Petri dish filled with solidified, non-nutritive 2% agar and several oat flakes. The sclerotium was inoculated with deionized water and colonize UV-sterilized oat flakes in a dark and humid 25 °C incubator for 48 h. Three oat flakes inoculated with plasmodial slime mold and four additional sterile oat flakes were placed on fixed locations of the map. The slime mold was incubated at 25 °C with no light for 72 h in a custom-built incubator to allow *P. polycephalum* to fully explore the map and create its network. Preliminary trials revealed 72 h to be ample time for full exploration of the 3D map by the slime mold. The need for the custom-built incubator came from the objective to take photographs of the entire map at 6-hour intervals over the period of each trial, but without exposing the maps to potential airborne fungal contaminants or to extended periods of light which both alter slime mold growth and behavior. In collaboration with two electrical engineering students at Loyola Marymount University, the incubator was built from off-the-shelf plastic bin and fitted with a Raspberry Pi camera and Graphical User Interface that enabled with a Python Script and an Optical Coherence Tomography System to capture photos of exploration of the slime mold over the map. Full description of the Slime Mold Incubator Camera System (SMICS) is detailed in (*McGrath et al., 2018*) (Fig. 1). From the maps generated at the end of the 72-hour exploration period, distances between oat flakes were measured using ImageJ software, and statistical analyses were conducted to compare the plasmodial slime mold's exploration networks with the analogous real-world urban roadway network in downtown Los Angeles (Fig. 2).

## RESULTS

Seventeen trials of the plasmodial slime mold's exploration on the proportional 3D topographical maps of the same region of Downtown Los Angeles were conducted (Fig. 3). The placement of oat flakes varied only slightly between the study's 17 replicates; however, the difference was not statistically significant (One Way ANOVA, $F = 0.04$, DF = 2, $\alpha = 0.05$, $p = 0.96$). The slime mold's overall exploration on the 3D maps of downtown Los Angeles was found to be shorter than human engineered, pre-existing roadway networks (Fig. 4). Slime mold networks were on 65.6 km $\pm$ 13.8 km (Average $\pm$ Standard Error) of equivalent length shorter than downtown Los Angeles' current urban roadway network. Furthermore, the number of oat flakes explored in each of the maps was suggested to positively affect the efficiency the slime mold had to explore larger distances. Statistical comparisons of the distances between pairs of real-world intersections and matching pairs of oat flakes revealed a significantly shorter pathway was created by the slime mold than

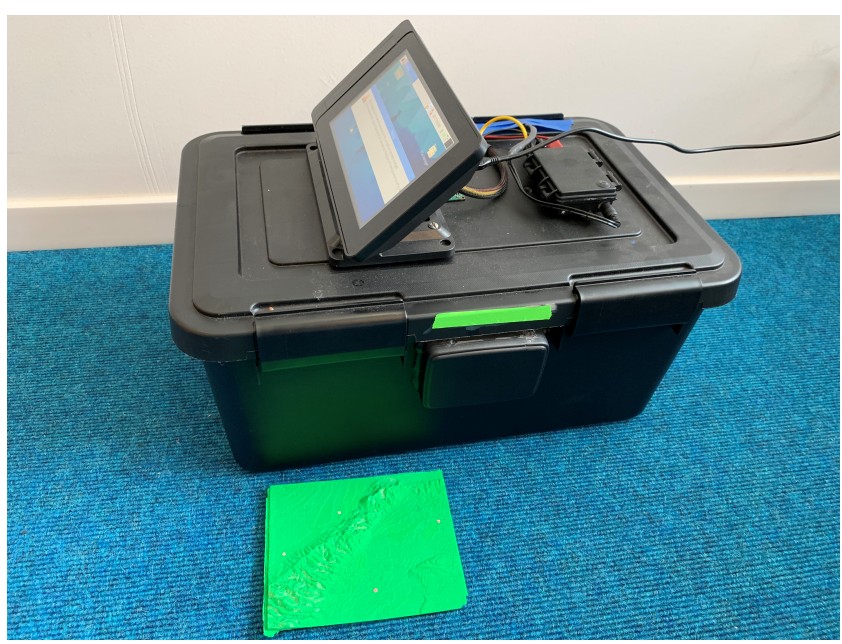

**Figure 1** Photograph of the custom-designed and built Slime Mold Incubator Camera System (SMICS) used to capture the exploration of the slime mold over the 3D printed maps with one of the 12 cm × 15 cm maps included for scale.

that of existing roadways ($t$-test for two independent groups $t = 5.37$, DF $= 218$, $\alpha = 0.05$, $p < 0.0001$).

## DISCUSSION

This study confirms that the plasmodial slime mold *P. polycephalum* can create networks comparable to real-world built networks as described in past 2D map studies (*Nakagaki, Yamada & Toth, 2001*; *Tero et al., 2010*). More so, this study illustrates that when presented with 3D printed maps, specifically those at the city-scale with real-world topographic features, the slime mold identifies pathways that are typically shorter and thus potentially more cost-effective than pre-existing city roadway networks. This is valuable because explorations such as these can reveal not only shorter pathways, but also variables such as network connectivity that if even slightly improved can reduce traffic commute time (*Levinson, 2012*).

The connectivity of the network provides insight into how paths are organized hierarchically. *Jiang (2009)* determined that a minority of roadways (20%) handle the majority of traffic (80%), with the top 1% of paths managing 20% of traffic flow. Thus, although the current study focused on a limited number of fixed locations (i.e., a fixed number of oat flake food sources) in the construction of the network, these locations were where major highways and roadways intersect and heavy traffic flows. Importantly, there is a cost associated with the lengths within a network that must span a large area and these lengths can dramatic influence the overall topography of the network (*Barthelemy,*

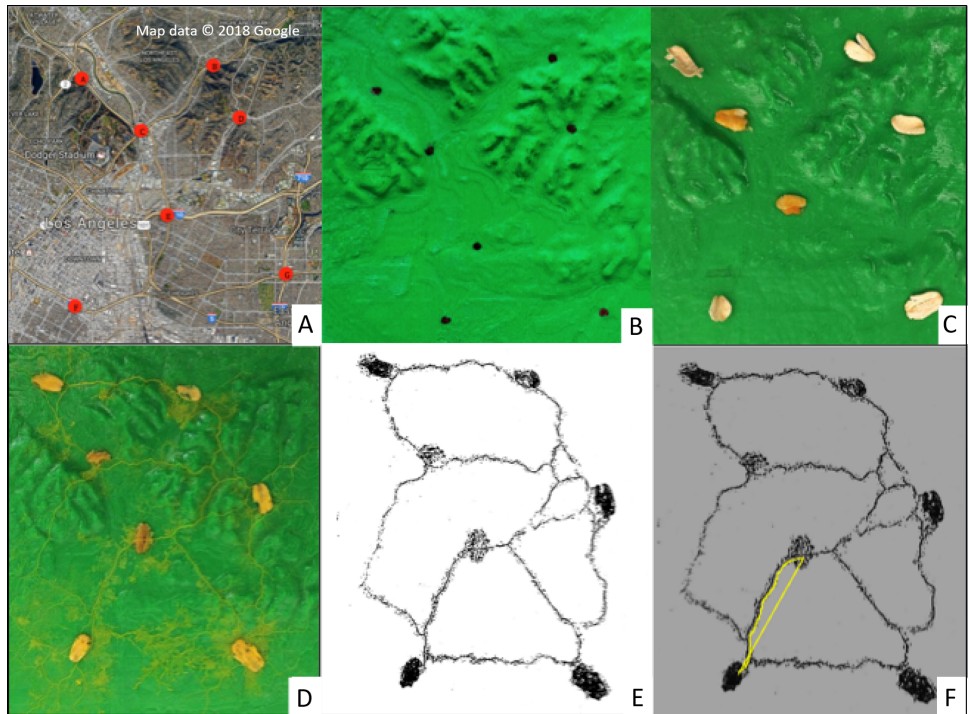

**Figure 2** Slime mold data generation sequence including (A) real-world aerial map with selected intersections of major roadways (red circles), (B) printed 3D map with topographic features and fixed locations (black dots), (C) representative 3D map with inoculated oat flakes at fixed locations, (D) resulting slime mold network (yellow threads) after 3-day incubation, (E) ImageJ software visualizing network connections, and (F) example length measurement of both freeform and direct distance between nodes using ImageJ tool.

*2011*). In an urban area, a single road length may be necessary to connect an isolated neighborhood or industrial facility to the rest of the network, consequently adding to the overall expansion and associated maintenance cost of the roadway network. A limitation of our experimental design was our focus on major roadway intersections and future work may examine such isolated, yet important elements of an urban landscape at a higher resolution, a task shown possible here with our examination using a scaled 3D map of only a 12 km × 15 km area of downtown Los Angeles.

Observed shorter slime mold pathways than real-world roadways may be influenced by how each network was created. Urban roadways are built over time, often in a non-linear progression with inherent restrictions including land ownership, economic resources, and a wide range of social, political, and historical considerations. Furthermore, the development of real-world roadways may be modeled by preferential attachment (*Barabasi & Albert, 1999*)—the concept that new nodes prefer to link to already well-connected network nodes. In an urban scenario, a new housing or commercial development may preferentially be connected directly to a population center, but at a net cost to the overall efficiency of the roadway network. In contrast, slime mold's exploration follows a minimalist strategy creating the most efficient links while curtailing resource expenditure (*Adamatzky*

## Real World Highways and Roads    Slime Mold Exploration Networks

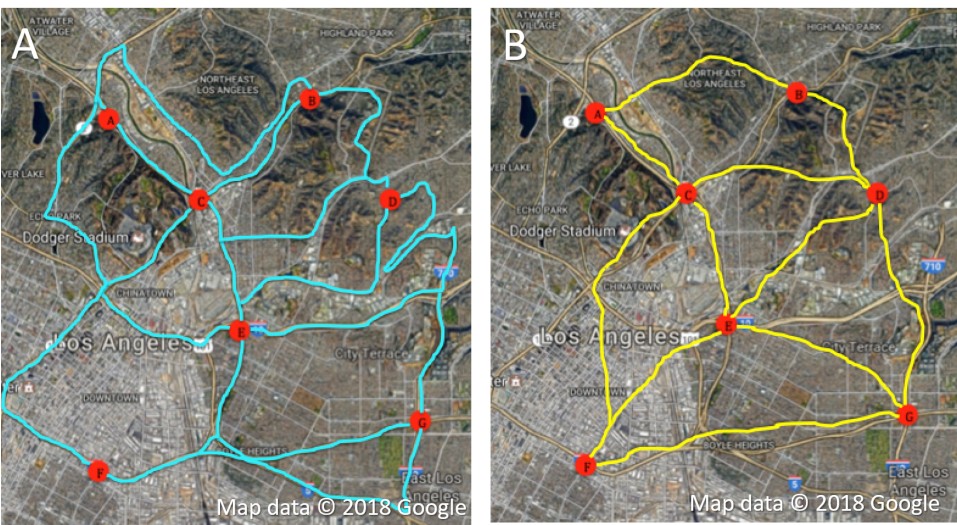

**Figure 3** **Comparison of slime mold exploration and existing road network.** Visual representation comparing the networks of the existing highway and road networks (A) to those that were created by the slime mold's exploration (B). This image also depicts the locations of the nodes of where the food sources (oat flakes) were placed on the 3D printed maps of Downtown Los Angeles. Image area approximately 12 km by 15 km. Map data ©2018 Google.

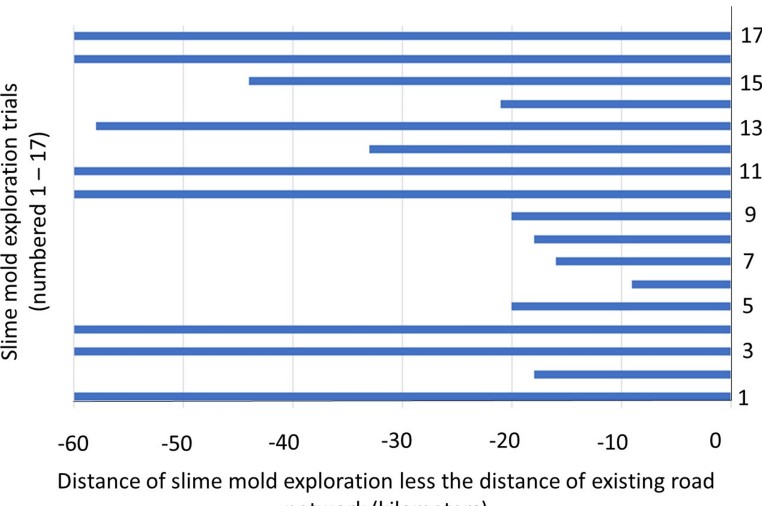

**Figure 4** **Calculated differences between slime mold and existing road network distances.** Difference in distance of the slime mold's entire exploration on 3D printed maps compared to that of existing highway networks.

*& De Oliveira, 2011*). Slime mold will abandon explored pathways that are no longer the most efficient for its metabolic function (achieved by leaving a non-living mat of transparent extracellular slime) (*Reid et al., 2012*), whereas urban roadways are not typically decommissioned or removed.

A high level of network similarity was qualitatively identified among the 17 72-hour iterations. These data suggest there are certain pathways across the landscape that are important to the overall connectivity and efficiency of the network. Recognizing which routes are important allows for resilience testing of the network in the event of a failure (*Quattrociocchi, Caldarelli & Scala, 2014*), such as in an earthquake, landslide, or anthropogenic disaster. Future studies may playout such scenarios by applying chemical deterrents such as sodium chloride (*Adamatzky & De Oliveira, 2011*) or a light beam (*Tero et al., 2010*) to inhibit the exploration of slime mold into disturbed areas, and then identifying the most efficient alternative routes.

Urbanization will continue in the coming decades with 68% of the world's population predicted to live in cities (*United Nations, 2018*). Urban growth often facilitates economic development but is also typically associated with the undesirable effects of increased traffic congestion and pollution (*Quaas & Smulders, 2018*). The application of emerging, yet appropriate technology may counter these unintended outcomes. For example, although between 1980 and 2000 the greater Los Angeles area grew by more than 40% and total vehicle mileage increased by 88%, ambient air quality including ozone, sulfur dioxide, carbon monoxide and lead all significantly improved with the application of vehicle emission reducing technology (*Kahn & Schwartz, 2008*). Likewise, we demonstrate here the potential for 3D printing technology paired with plasmodial slime mold can inform and guide network construction of urban roadways and potentially other infrastructure systems across real-world geographic features.

## CONCLUSIONS

Our data indicate slime mold presented with real-world topographic features can identify pathways shorter than current existing roadway networks, and the reoccurrence of certain links in multiple test iterations identify where key routes are in the urban roadway network. Given the multiple costs and benefits of urban roadway networks and how increased efficiency of networks can amplify the benefits, we argue for broader inclusion of biologically-inspired design, including the above described inexpensive and rapid approach of coupling slime mold and 3D print technology, in urban roadway discussions and planning.

## ACKNOWLEDGEMENTS

The authors thank Daniel McGrath and Stephen Board of the LMU Department of Electrical Engineering and Computer Science for the tremendous contribution in developing the SCIMS system for photo-documenting the growth and behavior of the slime mold. Thanks also to Dr. Michelle Yeung for technical support in printing the 3D maps, and the LMU

Summer Undergraduate Research Program for early support of S.D. D.W. thanks Dr. Marga Joaquin for logistical support.

### Funding
This project was funded with support from the Biology Department at Loyola Marymount University and the LMU Summer Undergraduate Research Program. The funders had no role in study design, data collection and analysis, decision to publish, or preparation of the manuscript.

### Grant Disclosures
The following grant information was disclosed by the authors:
Biology Department at Loyola Marymount University.
LMU Summer Undergraduate Research Program.

### Competing Interests
The authors declare there are no competing interests.

### Author Contributions

- Sophia Deen analyzed the data, conceived and designed the experiments, performed the experiments, prepared figures and/or tables, authored or reviewed drafts of the paper, and approved the final draft.
- Tatiana Kuzmenko conceived and designed the experiments, authored or reviewed drafts of the paper, and approved the final draft.
- Hossein Asghari conceived and designed the experiments, authored or reviewed drafts of the paper, construction of SMICS, and approved the final draft.
- Demian A. Willette analyzed the data, conceived and designed the experiments, authored or reviewed drafts of the paper, and approved the final draft.

### Data Availability
The raw base maps and distances and images from trials 1–17 are available in the Supplemental File.

### Supplemental Information
Supplemental information for this article can be found online at http://dx.doi.org/10.7717/peerj.8238#supplemental-information.

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
