# Peer review of "Investigating Los Angeles’ urban roadway network from a biologically-formed perspective"

_PeerJ, doi:10.7717/peerj.8238_

## Round 0.1 · original submission · Major Revisions

Both reviewers acknowledge the high potential of your work and think that it may provide an original contribution to the inclusion of biological systems in engineered systems. However, I concur with reviewer 2 that much effort should be made to expand the discussion on the experimental setup and to make more clear for the audience how your approach may contribute to solving real-world problems. Detailed remarks are reported in the reviewers' reports and should be taken into account when submitting the revised version of your manuscript.

·

Basic reporting

The quality of English language used throughout is of a generally high standard, although a proof read to pick up on minor typographical errors is necessary (especially on in-text citations).

A good sample of the literature is referenced, but as there are an enormous number of papers on the topic of slime mould navigation, the reviewer recommends a few more are referenced for completeness. Specifically, the authors will be interested in the following articles, which examine slime mould navigation across 3d-printed surfaces.
A. Adamatzky et al. (2014) Slime mould analogue models of space exploration and planet colonisation. JBIS, 67, pp.290-304
T. Dillon et al. (2019) Interspecies Urban Planning, Reimaging City Infrastructures with Slime Mould. In Adamatzky (ed.) Slime mould in Arts and Architecture. pp171-187. River Publishers.

The second reference, ln 204-205, is to an arXiv preprint. Please update to the final publication: https://doi.org/10.1142/8482

Figures are generally good, although the image quality seems rather low: possibly the review copy is compressed, however. It would also be beneficial to include a figure showing the authors' slime mould incubator.

The reviewer considers that the research question could have been made clearer in the introduction, although the concluding statements do make the purpose and outcome of this research explicit.

Experimental design

The experiments presented appear to have been soundly designed and conducted with suitable rigour. Please see the final comment in the previous section regarding clarifying the research question in the introduction.

A few details of the methods section should be amended for clarity, please see the following comments.
1- "Makerbot 3D printer" ln 112, please state the model
2- "sterilized" ln 121, sanitized would be a more appropriate description of the UV irradiation technique used
3- "2% agar" ln 122, the reviewer assumes that this was non-nutritive (AKA "water") agar and that it had been autoclaved?
4- "Purchased" ln 127, please state the supplier and, if available, the strain of P. polycephalum
5- "four additional sterile oat flakes", ln 131, the network density will be directly proportional to the number of food sources. Why were these four oat locations chosen? Do they correspond to population centres?
6- Following from point #5, the network characteristics will also be time dependent. Why was 72 hours chosen as the end-point of the experiment?
7- The dimensions of the printed topographic map and hence the scale reduction don't appear to be given.

Validity of the findings

The number of replicate experiments were suitable to make statistically-significant conclusions and the conclusions follow on from the results in a logical manner.

The following points should be addressed for clarity:
1- "average" ln 154, the standard deviation should be given too (the reviewer notes that the range is visible in fig 3)
2- "high level of networks similarity" ln 176, How was this assessed? If only qualitatively, this is fine, but needs to be stated. Also, typo: networks >> network

Additional comments

This was an interesting paper that develops the concept of transport network bioevaluation, whose novelty lies in comparing P. polycephalum navigation to extant road systems on 3D printed cityscapes with a z-topography. The reviewer recommends that the manuscript is suitable for publication if the points above and below are addressed.

General comments:
1- "single-celled" ln 59, P. polycephalum is not single-celled, but acellular, please correct.
2- "plasmodium tubes consist of actin-myosin fibres" ln 64, the wording here is somewhat misleading as the intra-tube structure of Physarum is complex, recommend rewording to better represent that tubes contain actin and myosin as the components to achieve locomotion and shuttle streaming (with a relevant citation).
3- Sentence starting ln 73, It should be made explicit here that some other studies (including 2 cited by the authors) have utilised 3d printed maps with z-topography, before proceeding to explain where the gap in knowledge is and how this paper addresses it.
4- Ln 86-97, The justification for using LA as a map doesn't seem to flow well. This paragraph appears to initially state that traffic in LA is bad, before stating it's highly ranked for automobile accessibility?

·

Basic reporting

Ti: Investigating Los Angeles' urban roadway network from biologically-formed perspective


This is an interesting report. In a broad sense, it attempts to further our understanding of how to incorporate biologically inspired systems in engineered systems such as the road system. In summary, it grows a network-forming slime mold into a 3D replica (2.4X) baited using oat flakes to recreate the human-engineered web of urban highways of Los Angeles, CA in the USA. They find a consistently shorted length of the network formed by the organism.

As I see it, while the work has enormous potential to develop interesting research, authors need to go a bit beyond the report presented here. The explicit development of a set of more ambitious research questions and hypotheses may help to frame the ms beyond a simple report of facts partially shown elsewhere presented here. Either in terms of energetics, by comparing energy costs in terms of the evolutionary adaptations or in terms of providing a more detailed description of the how these ~60 or so km less built by slime molds could really inform better planning for LA.

Experimental design

The experimental design is clear albeit extremely simple. I would have liked to see a more ambitious setup to deepen on the questions proposed.

Validity of the findings

While I find this piece of research very compelling I would have liked to see a deeper discussion, not only regarding the differences that may explain the difference between the empirical distribution of the road vs. slime mode network, but I also miss a more profound discussion on the experimental setup and the factors that may obscure this particular comparison and how it may contribute to use this type of approach for solving real-world problems. Particularly in the planning of urban systems and mobility. I truly believe that this type of contribution is a good first start, but we really need to make a large effort to convey this to a broader audience so that this kind of research may have a real impact on urban planners.

Additional comments

Ti: Investigating Los Angeles' urban roadway network from biologically-formed perspective


This is an interesting report. In a broad sense, it attempts to further our understanding of how to incorporate biologically inspired systems in engineered systems such as the road system. In summary, it grows a network-forming slime mold into a 3D replica (2.4X) baited using oat flakes to recreate the human-engineered web of urban highways of Los Angeles, CA in the USA. They find a consistently shorted length of the network formed by the organism.

As I see it, while the work has enormous potential to develop interesting research, authors need to go a bit beyond the report presented here. The explicit development of a set of more ambitious research questions and hypotheses may help to frame the ms beyond a simple report of facts partially shown elsewhere presented here. Either in terms of energetics, by comparing energy costs in terms of the evolutionary adaptations or in terms of providing a more detailed description of the how these ~60 or so km less built by slime molds could really inform better planning for LA.

While I find this piece of research very compelling I would have liked to see a deeper discussion, not only regarding the differences that may explain the difference between the empirical distribution of the road vs. slime mode network, but I also miss a more profound discussion on the experimental setup and the factors that may obscure this particular comparison and how it may contribute to use this type of approach for solving real-world problems. Particularly in the planning of urban systems and mobility. I truly believe that this type of contribution is a good first start, but we really need to make a large effort to convey this to a broader audience so that this kind of research may have a real impact on urban planners.

---

## Round 0.2 · Minor Revisions

A reviewer requires some further minor changes for better explaining the potential causes that may drive the differences revealed by the experiment.

·

Basic reporting

All comments addressed.

Experimental design

All comments addressed.

Validity of the findings

All comments addressed.

Additional comments

The reviewer is satisfied that all points highlighted have been satisfactorily addressed and hence recommends publication of the manuscript in its current state.

·

Basic reporting

I still find this research effort a very compelling and interesting addition to the literature. Authors have made an effort to improve the manuscript by including a new paragraph in the Discussion. However, the link between the results and the potential usage of the evidence gathered here is still dim to my understanding. I would like to see a deeper exploration of the specific reasons as to why slime mold outlines a smaller path-length in this particular landscape compared to the way LA has been built over time. There are evidently inherent restrictions in which the road network has been built that may be modeled using a preferential attachment approach but not discussed here. How do their experiments relate to such facts? Do they? Additionally, some other social restrictions may be in play here. They may point towards the role of redundancy in path elections in urban planning as opposed to other biological processes in the taxis of slime mold exploration. Adding this bit will certainly provide a stronger link between this interesting experiment and urban planning.

Experimental design

I have no issues with this section

Validity of the findings

The experiment presented here is very interesting and results potentially revealing. However, the link between the mere facts and the objective of making a parallel between how LA has been constructed and how slime mold explores the landscape can, to my understanding, be iimproved.

Additional comments

I recommend to further expand the discussion on the potential causes that may explain the differences revealed by the experiment.

---

## Round 0.3 · accepted · Accept

I think that the paragraph added in the revised version of the paper improves the link between the results obtained in the present study and the potential usage in real conditions.